# Surface Modification of Electrocatalyst for Optimal Adsorption of Reactants in Oxygen Evolution Reaction

Hong Soo Kim [1,†] , Hwapyong Kim [1,†] , Monica Claire Flores [1], Gyu-Seok Jung [2] and Su-Il In [1,*]

[1] Department of Energy Science & Engineering, Daegu Gyeongbuk Institute of Science & Technology (DGIST), Daegu 42988, Korea; cptmaos66@dgist.ac.kr (H.S.K.); khp911@dgist.ac.kr (H.K.); moniclaire@dgist.ac.kr (M.C.F.)

[2] Nextdoor Korean Medicine Clinic, Daegu 43018, Korea; blaze1122@naver.com

[*] Correspondence: insuil@dgist.ac.kr; Tel.: +82-053-785-6417

[†] These authors contributed equally to this work.

**Abstract:** Technological development after the industrial revolution has improved the quality of human life, but global energy consumption continues to increase due to population growth and the development of fossil fuels. Therefore, numerous studies have been conducted to develop sustainable long-term and renewable alternative energy sources. The anodic electrode, which is one of the two-electrode system components, is an essential element for effective energy production. In general, precious metal-based electrocatalysts show high OER reactions from the anodic electrode, but it is difficult to scale up due to their low abundance and high cost. To overcome these problems, transition metal-based anodic electrodes, which exhibit advantages with respect to their low cost and high catalytic activities, are in the spotlight nowadays. Among them, stainless steel is a material with a high ratio of transition metal components, i.e., Fe, Ni, and Cr, and has excellent corrosion resistance and low cost. However, stainless steel shows low electrochemical performance due to its slow sluggish kinetics and lack of active sites. In this study, we fabricated surface modified electrodes by two methods: (i) anodization and (ii) hydrogen peroxide ($H_2O_2$) immersion treatments. As a result of comparing the two methods, the change of the electrode surface and the electrochemical properties were not confirmed in the $H_2O_2$ immersion method. On the other hand, the porous electrode (PE) fabricated through electrochemical anodization shows a low charge transfer resistance ($R_{ct}$) and high OER activity due to its large surface area compared to the conventional electrode (CE). These results confirm that the synthesis process of $H_2O_2$ immersion is an unsuitable method for surface modification. In contrast, the PE fabricated by anodization can increase the OER activity by providing high adsorption of reactants through surface modification.

**Keywords:** electrocatalyst; oxygen evolution reaction; electrochemical anodization; stainless steel; surface modification

## 1. Introduction

As the first and second industrial revolutions took place, technological development through scientific innovation improved the quality of life, but worldwide energy consumption continues to increase due to population growth and the exploitation of fossil fuels [1–3]. Especially, most advanced technologies rely on fossil fuels, i.e., coal, natural gas, and petroleum, leading to an increase in atmospheric carbon dioxide ($CO_2$) concentration and polluting the global environment, which has a severe impact on the global ecosystem [4–6]. To regulate these problems, numerous climate laws have been enacted and enforced worldwide to reduce energy consumption and $CO_2$ emissions. Also, many people have a craving for long-term sustainable and renewable energy sources [2,7]. Therefore, various studies have been conducted to develop sustainable and renewable energy resources, such as alkaline water electrolysis [8,9], fuel cells [10,11], and metal–air batteries [12]. These technologies

are commonly composed of a two-electrode system, and among them, the performance of an anodic electrode is an essential factor for effective energy production.

In the anodic electrode, this part proceeds with the oxygen evolution reaction (OER, $2H_2O$ (l) $\rightarrow$ $4e^-$ + $4H^+$ (aq) + $O_2$ (g)) or oxidation of some chemical fuel, and the efficiency is affected by the OER kinetics of the electrode. In general, OER is a four electron–proton coupled reaction, and a high overpotential is required to overcome the kinetic barrier of OER [13–15]. Accordingly, various materials have been used to improve OER kinetics and stability of electrode, among which precious metal-based electrocatalysts showed excellent activity [13,14]. However, precious metal-based electrocatalysts are not practical for large-scale application and even restrict the development of electrochemical anodic electrodes due to low abundance and high price. Therefore, low cost and effective OER catalysts are essential for renewable energy sources. These alternative catalysts are being published through various studies [14–16]. Particularly, transition metal-based anodic electrodes, such as metal oxides [17], metal hydroxides [18], metal phosphides [19], and metal phosphates [20], exhibit advantages concerning their low cost and high catalytic activities. However, all candidates for transition metal-based electrocatalysts have inherent corrosion and oxidation susceptibility, limiting their use as OER anodic electrodes [14,21].

Stainless steel, composed of an alloy of transition metal, i.e., iron (Fe), nickel (Ni), and chromium (Cr), is an attractive material to use as a substrate for energy storage and electrocatalysis due to its excellent corrosion resistance and low cost [22–24]. However, stainless steel has low electrochemical performance due to its sluggish kinetics and lack of active sites [22,23,25]. Therefore, various strategies have been reported to enhance the properties of stainless steel-based anode for OER, for example, sulfurization treatment [26], cathodization treatment [27] of stainless steel foil and stainless steel fiber felt [28]. Our previous study also fabricated a novel porous electrode (PE) with a large surface area and low electrochemical resistance via the electrochemical anodization technique [29–33]. The anodization process has the advantage of being simple and easy to manufacture compared to other earlier reported technologies [34,35], which improves the electrochemical properties of stainless steel-based electrodes and is also suitable for mass production.

With an easy and simple process, the inventor Kou-Tsair SU reported the novel method of the porous layer through a 2011 US patent [36]. According to the patent, it is possible to make a porous electrode with a charge layer formed on the surface as well as a porous layer by immersion in a hydrogen peroxide ($H_2O_2$) solution. In this study, we investigated the synthesis process of this patent by attempting to reproduce their porous electrode soaked in $H_2O_2$ solution (hydrogen peroxide immersion electrode, HIE) and analyzed its physical and electrochemical characteristics with the PE. The electrochemical anodization and $H_2O_2$ immersion methods were set up as shown in Figure 1. Additionally, we compared the electrochemical OER activities of the PE and HIE to determine which method is more effective.

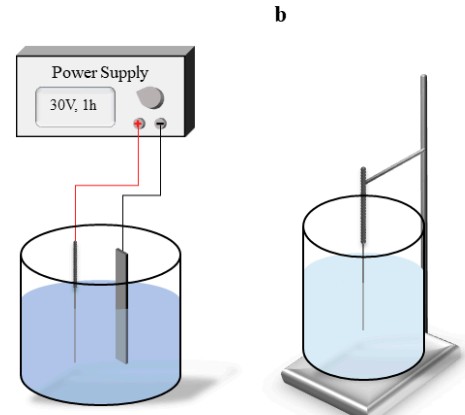

**Figure 1.** Schematic illustrations showing (**a**) electrochemical anodization setup used in preparing the PE and (**b**) hydrogen peroxide ($H_2O_2$) immersion setup used in preparing the HIE.

## 2. Results

### 2.1. Surface Morphology

Figure S1 shows the surface morphologies of the unprocessed conventional electrode (CE) and silicon-coated conventional electrode (Si-CE) obtained using a field emission scanning electron microscope (FE-SEM). Compared to the CE, which shows a smooth surface (Figure S1a), the PE, anodized using 0.3 wt.% $NH_4F$ and 2.0 vol.% DI water in ethylene glycol electrolyte, has a distinct micro/nano-scale porous structure on the surface, as shown in Figure 2a. Additionally, the cross-sectional image of the PE shows that the pores are conical in shape, with a width of 0.81–1.92 μm and a depth of 0.51–1.33 μm (Figure 2b). However, the electrode treatment with 10% concentration of $H_2O_2$ shows a similar surface appearance to the conventional electrode (Figure 2c). As claimed in the patent, the micropores on the surface of the processed electrode are not evident in the 10% concentrated hydrogen peroxide immersion electrode (10%-HIE) sample.

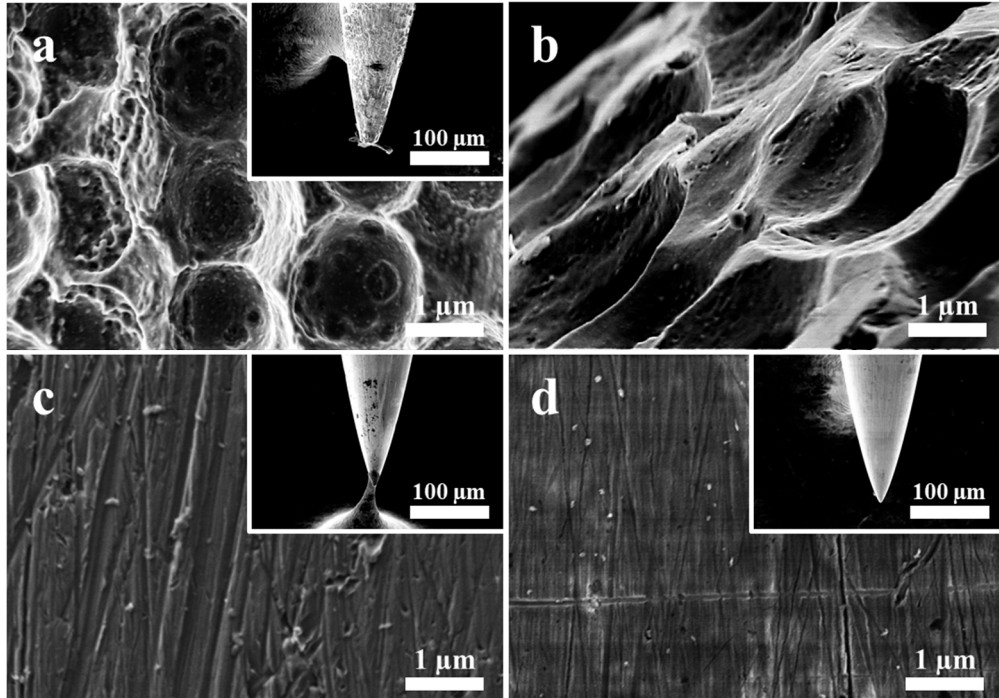

**Figure 2.** FE-SEM images of (**a**) porous electrode (PE), (**b**) cross-section of PE, (**c**) conventional electrode processed in 10% $H_2O_2$ immersion (10%-HIE) and (**d**) silicon-coated conventional electrode in 10% $H_2O_2$ immersion (Si-10%-HIE). Insets of (a,c,d) show electrode tips.

The synthesis method of HIEs was also conducted with varying concentrations of hydrogen peroxide solution, as shown in Figure S2. The possible pore production on the surface of the electrodes was investigated in N% $H_2O_2$ immersion (N = 10, 15, 20, 25 and 28%). Similar surface characteristics were observed at different concentrations, showing a negligible effect of the $H_2O_2$ solution to produce pores on the surface of the electrodes.

We further confirmed it by applying the same method as the synthesis process of HIEs to the silicon-coated conventional electrode (Si-CE). Compared to the 10%-HIE, the same concentration of $H_2O_2$ immersion silicon-coated electrode (10%-Si-HIE) does not show any pores on the surface and has almost similar smooth surfaces as the CE, as shown in Figure 2d. However, the 10%-Si-HIE has white spots on the surface, which is suggested to be caused by the dissolution of the silicon coating [37]. This chemical reaction is also observed when the silicon-coated conventional electrodes are processed in varying concentrations of $H_2O_2$ solution (Figure S3).

## 2.2. Electrichemical Characteristics

Electrochemical impedance spectroscopy (EIS) was used to investigate the electrical characteristics of CE, PE, N%-HIE and N%-Si-HIE electrode materials. Figure 3 shows the fitted Nyquist plots corresponding to EIS results for CEs, PEs, N%-HIEs and N%-Si-HIEs in saline solution. The curves exhibit the impedance of the working electrode attributed to the interaction of the charge transfer resistance ($R_{ct}$) and constant phase element (CPE) at the working electrode and electrolyte interface. It can be observed that the PE has a relatively smaller curve radius than the CE, which means that electrons flow with minimal resistance due to the low $R_{ct}$ between the working electrode interface and the saline solution. However, N%-HIEs and N%-Si-HIEs have larger curve radius than PE, and tend to decrease $R_{ct}$ with increasing concentration, but there is no significant change in charge resistance properties.

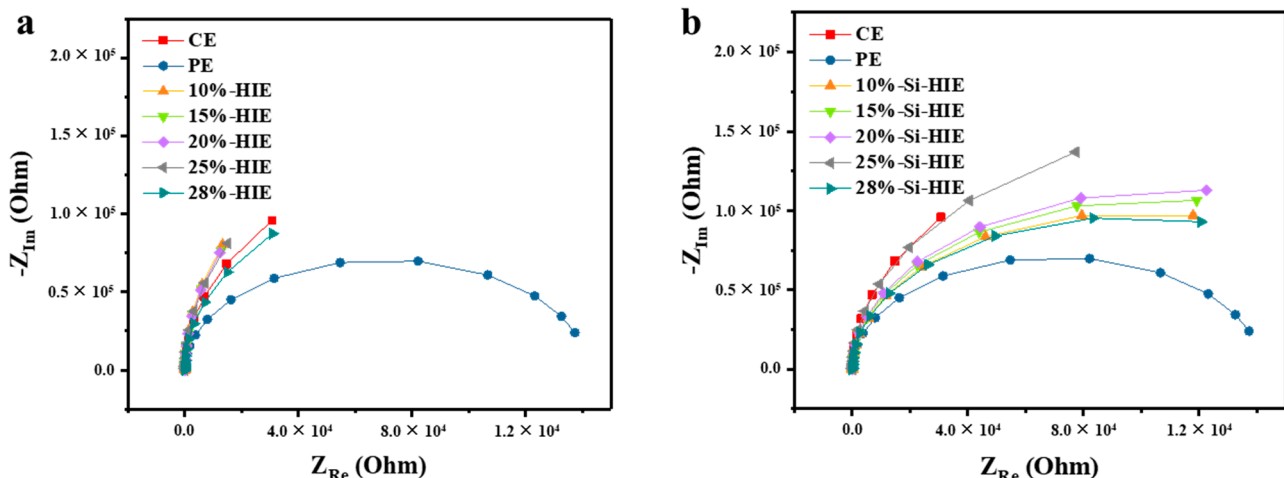

**Figure 3.** Fitted Nyquist plots corresponding to EIS results for (**a**) N%-HIE and (**b**) N%-Si-HIE as compared to CE and PE in saline solution (N = 10, 15, 20, 25 and 28%). Data obtained at 300 mHz–300 kHz, V = 30 mV.

## 2.3. Electrochemical Measurements for Oxygen Evolution Reaction (OER) Activity

The electrocatalytic activities of stainless steel electrodes were studied in alkaline solution for OER. To evaluate catalytic performance, linear sweep voltammetry (LSV) is usually performed in alkaline solution by applying over the thermodynamic potential of OER (>1.23 V vs. RHE) [38–40]. Figure 4a shows the polarization curves for the OER of the CE, PE, 28%-HIE, and 28%-Si-HIE. As the result, the PE shows the highest current density compared to the other stainless steel electrodes. However, the 28%-HIE and 28%-Si-HIE exhibit relatively low current density and electrochemical OER activity compared to the PE. Additionally, the 28%-HIE has a similar current density to the CE. Especially, for the 28%-Si-HIE, the current density is the lowest in the high applied voltage range, which is related to the high $R_{ct}$ due to the silicon coating on the electrode surface. Figure 4b shows the Tafel plots of various stainless steel electrodes. The PE has the smallest Tafel slope value of 67.6 mV dec$^{-1}$, suggesting the best catalytic electrochemical OER performance compared to other stainless steel-based electrodes. The other stainless steel electrodes show similar Tafel slopes, implying their similar OER kinetics.

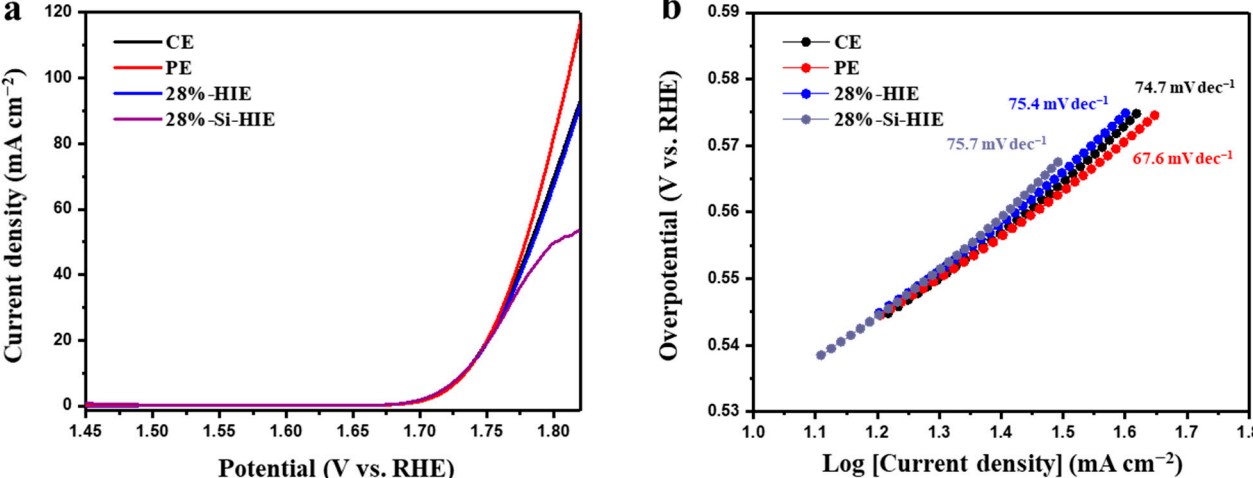

**Figure 4.** (**a**) Polarization curves for electrochemical OER activity and (**b**) the corresponding Tafel plots of CE, PE, 28%-HIE and 28%-Si-HIE.

## 3. Discussion

### 3.1. Discussion of $H_2O_2$ Immersion Electrodes (HIEs) Based on Patent

The patent provided schematic drawings and an electron microscope photograph of the porous electrode surface in Figures S4 and S5. As described by the patent document, the electrode portion (11) has an electrode body (111) surrouded by a microporous layer and covered by a charge layer (Figure S4a). The porous layer (112), as schematically illustrated in Figure S4b, was formed by 20–1000 nm sized micropores (113) with a thickness of 50–5000 nm and a negative charge layer (114) suggested to be formed by hydroxyl groups (OH⁻) [36].

The electron microscope image of the porous electrode, wherein the specks of light-colored spots were claimed to be the micropores, and the dark background is the electrode body, is shown in Figure S5. The patent invention stated that the pore diameters could be in nanoscale after processing [36]. However, the patent inventor has not disclosed adequate information on the image parameters, such as the magnification or scale of the photograph. Thus, the actual measurements of the micropores are still indeterminate.

In addition, the patent invention asserted that a negative charge layer formed on the surface of the electrode has a dielectronic constant measured in the range of 2 to 110 F m⁻¹ [36]. According to the patent, the negative charge layer provides increased electrochemical reaction because of the surface potential difference on the electrode portion.

### 3.2. Discussion of Results

The PE, which was fabricated through electrochemical anodization, clearly shows the hierarchical micro/nano-scale pores on the surface of the electrode (Figure 2a,b), which is the same as previously reported [29–33]. In addition, due to the large surface area of the PE, it has low electrochemical impedance and $R_{ct}$, as shown in Figure 3, which can cause high adsorption and electrochemical reaction with reactants on the electrode surface. Thus, the PE shows a higher current density and lower Tafel slope of electrochemical OER activity in Figure 4, which is suitable for improving the OER performance of stainless steel-based electrodes through the anodization process.

Meanwhile, in Figure 2c and Figure S2, N%-HIE samples (N = 10, 15, 20, 25 and 28%) show a more similar surface morphology to the unprocessed CE; that is, they do not show a noticeable porous structure. The analysis shows that the fabrication method discussed by the patent is not reproducible of pore production on the surface of the electrode. In addition, since the description of the stainless steel electrode used in the patent was insufficient, the possibility of producing a porous structure on the surface of a Si-CE was also tested using the fabrication method for the patent. Figure 2d and Figure S3 show

that the N%-Si-HIE samples have not shown pores on their surface after processing in $H_2O_2$ solution, and only white spots caused by the dissolution of the silicon coating are present on the electrode surface [37]. Moreover, as the result of the EIS analysis, the HIE and Si-HIE samples do not show the significant change in the electrochemical impedance and $R_{ct}$ (Figure 3), which does not clearly explain the electron transfer properties of the negative charge layer on the surface described in the patent [36,41]. Therefore, the HIE and Si-HIE samples exhibit low current density and a high Tafel slope of electrochemical OER activity in Figure 4, suggesting that the $H_2O_2$ immersion process is not befitting for improving the OER performance.

## 4. Materials and Methods

### 4.1. Preparation of Porous Electrodes (PEs)

Conventional stainless steel type 304 (SUS304) electrodes (6.0 cm in length, 0.03 cm in diameter, and without silicon oil coating), see Figure S1a, were obtained from Dong bang Acupuncture Inc., Boryeongsi, Korea. Before anodization, the electrodes were consecutively cleaned with acetone, ethanol, and deionized (DI) water. Anodization of SUS304 electrodes was performed using a two-electrode cell, with the electrode as the anode and carbon paper (Carbon and Fuel cell (CNL), SGL 39BC, Seoul, Korea, 5 cm × 1 cm × 0.325 mm) as the cathode, see Figure 1a. Anodization was carried out for an hour at 30 V, using an electrolyte comprised of 0.3 wt.% (weight percent) $NH_4F$ (98%, American Chemical Society (ACS) reagent, Sigma-Aldrich, St. Louis, MO, USA) and 2.0 vol.% (volume percent) DI water in ethylene glycol (Extra Pure, Daejung, Siheung-si, Korea). After anodization, the electrode was successively rinsed with acetone, ethanol, and DI water and dried in a flowing stream of nitrogen.

### 4.2. Preparation of $H_2O_2$ Immersion Electrodes (HIEs) and $H_2O_2$ Immersion Silicon-Coated Electrodes (Si-HIEs)

This method was adopted from US patent 2011/0245856A1 [36]. Two types of conventional stainless steel type 304 (SUS304) electrodes (6.0 cm length, 0.03 cm diameter, with and without silicon oil coating), see Figure S1, were obtained from Dong bang Acupuncture Inc., Boryeongsi, Korea. Before surface modification, the electrodes were sequentially cleaned with acetone, ethanol and finally rinsed with DI water. Then, the surface of the SUS304 electrode was modified by immersing the electrode in $H_2O_2$ solution (Extra pure grade, Duksan, Seoul, Korea), as shown in Figure 1b. This immersion process was carried out for 10 h, using different N% concentration (N = 10, 15, 20, 25, and 28%) $H_2O_2$ solution. After this, the electrodes were dried in a flowing stream of nitrogen.

### 4.3. Characterization of Electrode Samples

Surface morphologies were evaluated using a Field Emission Scanning Electron Microscope (FE-SEM, Hitachi S-4800, Tokyo, Japan) operating at 3 kV, 10 μA. Electrochemical impedance spectroscopy (EIS) spectra were obtained using a VSP potentiostat (Bio Logic, Seyssinet-Pariset, France) three-electrode workstation with platinum (Pt) wire as the counter electrode, Ag/AgCl electrode as the reference electrode and test samples (PEs, HIEs, and Si-PEs) as the working electrode. The above-mentioned three electrodes were immersed in saline solution (0.9 g NaCl in 100mL DI water) purchased from JW-Pharma, Seoul, Korea. The EC Lab software was used to operate the system from 300 mHz to 300 kHz.

### 4.4. Electrochemical Measurements for Oxygen Evolution Reaction (OER) Activity

The oxygen evolution reaction activity was measured by linear sweep voltammetry (LSV) using 1M KOH (85%, Extra Pure, Daejung, Siheung-si, Korea). Linear sweep voltammetry was conducted in a conventional three-electrode system using VSP potentiostat (Bio Logic, Seyssinet-Pariset, France) with a scan rate of 5 mV s$^{-1}$. A saturated calomel electrode (SCE) was used as a reference electrode, and Pt wire served as a counter electrode.

The working electrodes were CE, PE, 28%-HIE, and 28%-Si-HIE as prepared. The catalytic activity for OER is evaluated from +0.35 to +0.75 V vs. SCE. This potential range value versus saturated calomel reference electrode was converted to the potential value versus reversible hydrogen electrode according to Equation (1) [42].

$$E_{RHE} = E_{SCE} + 0.244 + 0.0591 \times pH \tag{1}$$

The Tafel plot is modeled by the empirical Tafel Equation (2). Where η is the overpotential, a is the intercept relative to the exchange current density, b is the Tafel slope, and j is the current density [43].

$$\eta = a + b \times \log j \tag{2}$$

## 5. Conclusions

We reproduced and analyzed two surface modification processes: electrochemical anodization and the $H_2O_2$ immersion process. The PE fabricated through anodization has a uniform porous structure, which allows it to have a high surface area and electrochemical properties. In particular, the large surface area through the porous structure of PE can induce high adsorption between the reactant and the electrode surface during OER, resulting in high OER activity. However, the HIE and Si-HIE through the $H_2O_2$ immersion process claimed in the patent do not show any surface change, low electrochemical properties and OER activity. These results show that the content claimed by the patent is different, and suggest it is necessary to objectively check the facts of the data presented in the patent and paper. We hope that this paper will encourage researchers to explore further advances in electrocatalysis and electrode materials.

**Supplementary Materials:** The following are available online at https://www.mdpi.com/article/10.3390/catal11060717/s1, Figure S1: Surface and electrode tip FE-SEM images of (a) conventional electrode (CE) and (b) silicon-coated conventional electrode (Si-CE) as obtained from Dong bang Acupuncture Inc., Boryeongsi, Korea, Figure S2. Surface and electrode tip FE-SEM images of conventional electrodes processed in (a) 15%, (b) 20%, (c) 25% and (d) 28% $H_2O_2$ immersion (N%-HIEs), Figure S3. Surface and electrode tip FE-SEM images of silicon-coated conventional electrodes processed in (a) 15%, (b) 20%, (c) 25% and (d) 28% $H_2O_2$ immersion (N%-Si-HIEs), Figure S4: Detailed description drawings of the patent in (a) sectional view of the structure of the porous electrode and (b) magnified view of the porous layer, Figure S5: Scanning electron microscope (SEM, JEOL, JSM 6500, 15 kV acceleration voltage, $9.63 \times 10^{-5}$ Pa vacuum) image of the porous electrode surface replicated from the patent document.

**Author Contributions:** Conceptualization, S.-I.I.; data curation, H.S.K. and H.K.; formal analysis, H.S.K.; funding acquisition, S.-I.I.; investigation, H.S.K.; methodology, S.-I.I. and H.S.K.; project administration, S.-I.I.; resources, S.-I.I.; supervision, S.-I.I.; writing—original draft, H.S.K. and M.C.F.; writing—review and editing, H.S.K., H.K., G.-S.J., M.C.F. and S.-I.I. All authors have read and agreed to the published version of the manuscript.

**Funding:** This research was supported by a grant from the Korea Health Technology R&D Project through the Korea Health Industry Development Institute (KHIDI), funded by the Ministry of Health and Welfare, Korea, grant number HI19C0506.

**Institutional Review Board Statement:** Not applicable.

**Informed Consent Statement:** Not applicable.

**Data Availability Statement:** All data are shown in this article and the Supplementary Materials.

**Conflicts of Interest:** The authors declare no conflict of interest.

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
