# Peer review of "Surface Modification of Electrocatalyst for Optimal Adsorption of Reactants in Oxygen Evolution Reaction"

_catalysts, doi:10.3390/catal11060717_

Round 1

Reviewer 1 Report

In this paper the authors discuss on some characterization and experimental verifications on electrodes produced via anodization process following a procedure reported in a patent recorded in 2011.

Stainless-steel type electrodes and Si-coated electrodes are treated in a similar way with the aim to modify their morphology.

actually it seems that only in a single case the morphology can be modified with some increment in roughness / porosity. In all the other cases no significant modifications can be observed. The electrodes were then tested by means of EIS. Also here no significant results can be observed

I really cannot see any interesting scientific result in this manuscript. the presentation lacks of details and discussion, the english grammar is bad and it sounds just like some tests related to a patent 10 years old

I cannot recommend the publication.

Reviewer 2 Report

The work “Surface Modification of Electrolyte for Optimal Adsorption of Reactants in OER Reaction” by Hong Soo Kim et al. study the reproducibility for the fabrication of hydrogen peroxide immersed electrode and compare the bioelectrode characterization of porous bioelectrode synthesized by a electrochemical anodization process. Authors report in the study of surface morphology comparison using FE-SEM that there were no noticeable pores observed on the surface of HIEs compared to the easily distinct porous structure of porous bioelectrodes. Authors claim that their study confirms that the electrochemical anodization process is the first effective fabrication method for elaboration porous electrodes for enhanced electrochemical OER performance.

The paper reads well and the results of studies are clear and complete. The work is timely for the Catalysts community.

Reviewer 3 Report

In this work, the authors tested the reproducibility for the fabrication of HIE and compared the bioelectrode characterization. They cocluded that electrochemical anodization process is the first effective fabrication method to obtain porous electrodes for enhanced electrochemical OER performance. The content of the manuscript is of little novelity and I do not see this work is a complete piece of work that need to publish. I see this work as a observation not a complete work. Author should perform OER analysis and compare with well established protocoal. 

Round 2

Reviewer 1 Report

in the previous submission I rejected this paper due to some significant criticisms. 

I appreciate the effort to improve the manuscript in this new submission, anyway, in my opinion the results should be reported in a different way in order to make the paper suitable for publication.

 I think that the "bad" results should be moved to a Supporting information file, while the main text should discuss in details only the positive improvements in the investigated process. moreover, I recommend that the SEM micrographs illustrating the morphology of the samples should be done at higher magnification in order to better show the differences (now it is really impossible to see difference among the samples)

With a new arrangement of the paper I can reconsider the decision for publication

Reviewer 3 Report

The author modified the MS and included additional analysis and experiments. I am happy to accept this version in the current form.

Thanks

Round 3

Reviewer 1 Report

The authors did a major effort to improve the manuscript according to my comments. 

I can now recommend the publication